# OpenReview forum: "Unifying Stable Optimization and Reference Regularization in RLHF"
_ICLR.cc/2026/Conference — ICLR 2026 Poster_

### Official Review · Reviewer_4Fmj · 2025-10-18

**Soundness:** 3
**Presentation:** 3
**Contribution:** 3
**Rating:** 6
**Confidence:** 2

**Summary:**

This work proposes a unified regularization approach with weights that explicitly balances the objectives of preventing reward hacking and maintaining stable policy updates in RLHF.

**Strengths:**

The authors found a reasonable limitation of the existing approach to solve reward hacking and maintain stable policy updates, i.e., the conflict of two regularizers that pull the trained policy towards the reference policy and previous-step policy respectively. This well motivates the proposed dual-KL approaches, with the novel and straightfoward idea of combination of the two regularizers, which are presented clearly. The experiments look comprehensive to me.

**Weaknesses:**

A few clarity issues as shown in the following questions.

**Questions:**

(1) Is Eq. (2) PPO for RL not RLHF, since there is no KL penalty to $\pi _ {\rm ref}$? The equation in Section 4.1 seems to transform the KL penalty in Eq. (1) into hard constraint $KL<\epsilon$, yes?

(2) "Empirical Validation" in Section 4.1 involves two dual-KL variants. Why not move "Empirical Validation" to after introducing two dual-KL variants?

(3) Your dual-KL (Eq. 3) looks like a special case of [1] with 2 references, what are your differences and additional contributions?
[1] Gholamali Aminian, Amir R Asadi, Idan Shenfeld, and Youssef Mroueh. Theoretical analysis of kl-regularized rlhf with multiple reference models. ArXiv:2502.01203, 2025.

(4) In proposition 4.1, $\log\pi_{\theta}(y|x)=\alpha\log\pi_0(y|x)+(1-\alpha)\log\pi_t(y|x)+C(x)$ with a constant normalization $C(x)$ is suggested to ensure policies summing up to 1.

(5) What are the evaluation metrics of MT Bench in Table 3?

(6) In Figure 5c, could you explain more about EOS-missing rate, and why $\alpha=0$ increases EOS-missing rate? Is it convenient to add LC-win rate?

(7) In Algorithm 1, what are $\mu_A$ and $\sigma_A$? What are the meanings of the two weights $w _ {\rm reg}^i$ and $w _ {\rm adv}^i$ and how are they computed?

---

> ### Author Response · Authors · 2025-11-23
>
> We greatly appreciate the reviewer's positive evaluation and constructive suggestions. We will reflect and update based on your comments in our revisions, and address each of these valuable points as follows:
>
> ### Q1: Clarification on PPO for RLHF and PPO-Align
> We appreciate the opportunity to clarify this distinction. While Eq. (2) represents the standard PPO objective for RLHF, the KL penalty with respect to the reference policy is incorporated into the reward definition (as shown in Line 152), which then yields the complete PPO for RLHF objective.
>
> In contrast, PPO-Align (Section 4.1) is derived via: (1) explicitly transforming the policy ratio clipping term $\text{clip}(\frac{\pi\_\theta}{\pi\_{t}}, 1\pm \epsilon)$ back to a hard-constrained formulation; (2) move the KL penalty term out from the reward/advantage definition as an explicit regularization term. This reformulation reveals that standard policy ratio clipping achieves stable optimization through mechanisms operating outside the stated optimization objective, which is the root of the overly restrictive regularization framework.
>
> ### Q2: Reordering subsection in Method
> Thank you for this suggestion. We have restructured the Method section accordingly in our revision.
>
> ### Q3: Comparing with multi-reference RLHF
> We acknowledge this concurrent work and cite it in our related work section. There are three fundamental differences:
> 1. **Motivation:** Multi-reference RLHF aims to incorporate multiple reference targets to ensure diversity in alignment, whereas our work addresses the inherent conflict between stable optimization and reference regularization in the standard RLHF framework.
> 2. **Optimization objective:** We employ a weighted regression loss, while multi-reference RLHF optimizes the classic reward-maximizing objective.
> 3. **Dynamic vs. static regularization:** Our regularization incorporates a dynamic component (π_t) that becomes progressively more aligned with preferences during training, whereas all reference policies in multi-reference RLHF remain fixed throughout training.
>
> ### Q4: The normalizing factor
> We appreciate this feedback and have incorporated appropriate clarifications regarding the normalizing factor in our revision.
>
> ### Q5: The evaluation metric of MT-Bench
> MT-Bench consists of 80 multi-turn questions covering diverse topics including mathematics, reasoning, conversation, and more. Each response is evaluated by GPT models (we use both GPT-4 and GPT-4-Turbo in our experiments), which assign a score from 1-10 for each turn. The reported score is the average across all turns and questions. Following your feedback, we have added a detailed benchmark description in the appendix.
>
>
> ### Q6: Clarification on EOS-missing rate
> The EOS-missing rate is calculated as the percentage of responses that terminate improperly without an EOS token. When α=0, regularization relies entirely on the current sampling policy $\pi\_t$, which effectively reduces to optimizing the RLHF objective without reference regularization. Since LLM-based reward models often exhibit length bias (preferring excessively long responses), the policy without reference regularization exploits this bias, leading to a higher EOS-missing rate.
>
> We report reward, EOS-missing rate, and response length rather than LC (length-controlled win rate) because these metrics more directly demonstrate the exploitative behavior of DAR under low α values. LC can more appropriately reflect the trade-off between length and quality, while here we provide clearer insights into the length exploitation phenomenon.
>
> ### Q7: Mean, Std, w_reg, and w_adv
> Thank you for requesting this clarification. These refer to the batch-based mean and standard deviation of the advantages, where w_reg and w_adv correspond to the regularization weight and advantage weight in Eq. (4), respectively. We have improved the clarity of these notations in our revision.
>
> ---
>
> We sincerely thank your constructive feedback, which has helped us improve the quality and clarity of our paper.

---

> > ### Comment · Reviewer_4Fmj · 2025-11-27
> > **Solved. I keep my rating=6.**
> >
> > The authors solved my questions. I keep my rating=6.
> > Reviewer 4Fmj

---

> > > ### Author Response · Authors · 2025-11-27
> > >
> > > Thank you for acknowledging that the rebuttal has resolved your concerns. We are grateful for your time and expertise in evaluating this submission, which indeed helped strengthen the paper.

---

### Official Review · Reviewer_GWzK · 2025-10-20

**Soundness:** 3
**Presentation:** 3
**Contribution:** 3
**Rating:** 4
**Confidence:** 3

**Summary:**

The paper proposes a *dual-KL regularization* approach that aims to jointly address two RLHF pain points: (i) preventing reward hacking via reference regularization and (ii) ensuring stable optimization via trust-region style control. The method first studies a constrained “PPO-Align” objective and then derives a *weighted dual-KL* objective that yields an interpretable, weighted SFT (DAR) loss by effectively interpolating between the initialization policy $\pi_{0}$ and the current policy $\pi_{t}$ in log space. Experiments on TL;DR, Anthropic Helpfulness, and Harmlessness report improved win rates (Figure 3/Table 2), with evaluations judged by GPT-4 Turbo. Figure 1 provides the conceptual motivation for expanding the search region beyond the intersection of the trust regions to balance stability and reference adherence.

**Strengths:**

- **Timely problem framing.** The paper clearly motivates the dual goals of stabilizing policy updates while constraining drift from a reference policy, and argues for a unified objective rather than separate mechanisms.
- **Empirical gains.** Across three benchmarks, DAR shows strong win rates against online RLHF (e.g., PPO, GRPO, RLOO) and online DAP baselines; curves in Figure 3 and the summary in Table 2 support the claim.
- **Implementation clarity intent.** The paper points to code/supplement details, which is important given the number of components interacting (policy, judge, datasets, ablations).

**Weaknesses:**

- In the discussion of PPO stability, the classical TRPO/PPO literature typically regularizes with a KL of the form $D_{\mathrm{KL}}\left[\pi_{\text{old}}\||\pi_{\theta}\right]$, see Schulman et al. (2015, TRPO) and Schulman et al. (2017, PPO). By contrast, the paper’s *PPO-Align* (Sec. 4.1) constrains with $D_{\mathrm{KL}}\left[\pi_{\theta}\||\pi_{t}\right]$ and penalizes $D_{\mathrm{KL}}\left[\pi_{\theta} \||\pi_{0}\right]$. The rationale for changing directions relative to the classical trust-region view is not made explicit. This makes it hard to judge whether the final dual-KL choice is a principled departure or just a convenient variant.
- Figure 3/Table 2 use GPT-4 Turbo as the judge and Qwen2-72B-Instruct as the LLM annotator. Results would be stronger with a stronger contemporaneous judge (e.g., GPT-5) and annotator (Qwen3) for additional validation.
- While DAR (weighted SFT) is argued to be stable, quantitative stability analyses for the *dual-PPO* path are sparse (e.g., per-iteration KL to $\pi_{0}$ and $\pi_{t}$, gradient-norm/entropy trends, collapse rates across seeds).
- Most results focus on Qwen2 and Llama-3.1 settings; adding a recent backbone (e.g., Qwen3) would better probe portability.
- *(Minor)* Colors/encodings are hard to parse quickly; the caption should explicitly map colors/shapes to policies/regions and call out what “search expansion” specifically denotes.

**Questions:**

- Given the centrality of Table 2, the community would benefit from end-to-end scripts (prompts, judge configs, seeds, filtering) in the supplement to exactly reproduce those numbers. Can we provide that in supplement?
- Would a *mixed-direction* dual-KL be preferable on theory/empirics—e.g., a mode-covering  term $D_{\mathrm{KL}}\left[\pi_{\theta}\||\pi_{0}\right]$ *and* a mode-seeking term around the behavior/current policy $D_{\mathrm{KL}}\left[\pi_{t}\||\pi_{\theta}\right]$?
- Could you report training stability metrics such as gradient-norm statistics, entropy, and seed-wise variance for dual-PPO?
- How do conclusions change with newer backbones (e.g., Qwen3) or different reward models?

---

> ### Author Response · Authors · 2025-11-23
>
> We thank the reviewer for the positive feedback on our motivation and implementation transparency. We have carefully considered your concerns and provide detailed responses below:
>
> ### Q1: Forward KL in TRPO/PPO vs. Reverse KL in Our Objective. Would a mixed-direction dual-KL be preferable?
>
> We acknowledge the reviewer's observation about KL divergence formulations. We appreciate for this opportunity to clarify:
>
> **Correction on PPO-Align:** There is a typo in the PPO-Align formulation. It should be constrained with Forward KL $\mathbb{D}\_\text{KL}(\pi\_t||\pi\_\theta)$ and penalized with Reverse KL $\mathbb{D}\_\text{KL}(\pi\_\theta||\pi\_0)$.
>
> **Justification for Reverse KL:** Using Reverse KL for stable optimization is a crucial component in deriving our weighted regression algorithm, which follows the approach in our prior work (AWR [1]). The Forward KL between the current policy and learning policy ensures monotonic policy improvement when using the current policy as a valid approximation to the learning policy. This motivation is consistent with the stability objectives in TRPO/PPO, justifying our use of Reverse KL for stable optimization.
>
> **Empirical analysis on Forward KL vs Reverse KL:** To ground the difference between Forward KL (mode-covering) and Reverse KL (mode-seeking), we implemented Dual-Mix-PPO using mixed KL where using Forward KL for stability, Reverse KL for reference penalty. Note that Mix-DAR is infeasible because the weighted SFT loss requires the stability constraint strictly in Forward KL form.
>
> Below we report mean rewards and standard deviations across three seeds. For Dual-Mix-PPO, we tuned $\alpha \in$ [**0.1**, 0.3, 0.5], $\beta \in$ [0.01, **0.02**, 0.03], with selected values in bold:
> |Mean Reward|1/6 Epoch|2/6 Epoch|3/6 Epoch|4/6 Epoch|5/6 Epoch|1 Epoch|
> |:-|:-:|:-:|:-:|:-:|:-:|:-:|
> |Dual-Mix-PPO|6.377±0.100|7.675±0.774|7.737±0.828|7.870±0.542|8.044±0.322|8.170±0.196|
> |Dual-PPO|6.683±1.110|8.131±0.726|8.417±0.515|8.666±0.285|8.804±0.409|9.028±0.226|
>
> These results (together with the added Figure 8 in our revised manuscript) indicate that the mode-covering approach (Forward KL) tends to be more conservative than the mode-seeking variant (Reverse KL). This conservatism leads to suboptimal learning performance, as the mode-covering penalty heavily restricts the learning policy's exploration for better responses.
>
>
> ### Q2: Validation with Stronger Judge, Annotator? How do conclusions change with newer backbones (e.g., Qwen3) or different reward models?
> We conducted additional experiments using:
> * Base model: Qwen-2.5-7B (we selected this over Qwen-3-7B because Qwen-3 models are instruction fine-tuned)
> * LLM annotator: Qwen-3-32B (as there is no 72B variant available from Qwen-3)
> * Preference judge: GPT-5.1
>
> The results below compare DAR against online RLHF methods, demonstrating that DAR consistently outperforms baseline methods across all datasets over three seeds:
> |Ref Win%|TL;DR|Helpful|Harmless|Mean|
> |:-|:-:|:-:|:-:|:-:|
> |PPO|60.70%±5.76%|61.73%±0.95%|72.50%±3.76%|64.98%|
> |RLOO|68.80%±1.48%|84.52%±0.54%|76.63%±1.64%|76.65%|
> |GRPO|68.27%±1.68%|77.72%±0.76%|79.63%±1.70%|75.21%|
> |**DAR**|**80.07%±0.64%**|**86.46%±0.54%**|**81.28%±0.40%**|**82.60%**|
>
> ### Q3: Quantitative Stability Analysis for DAR vs. Dual-PPO
>
> Our claim of improved learning stability for DAR is supported by Figures 5(a) and 5(b), which show that Dual-PPO's reward curves (averaged over three seeds) exhibit significantly larger confidence intervals compared to DAR, indicating higher variance across training runs.
>
> For quantitative validation, we computed mean rewards and standard deviations at each checkpoint:
>
> **Helpful-base:**
> |Mean Reward|1/6 Epoch|2/6 Epoch|3/6 Epoch|4/6 Epoch|5/6 Epoch|1 Epoch|
> |:-|:-:|:-:|:-:|:-:|:-:|:-:|
> |Dual-PPO|6.683±1.110|8.131±0.726|8.417±0.515|8.666±0.285|8.804±0.409|9.028±0.226|
> |DAR|7.574±0.089|8.676±0.063|8.864±0.087|9.260±0.026|9.215±0.080|9.370±0.007|
>
> **TL;DR:**
> |Mean Reward|1/6 Epoch|2/6 Epoch|3/6 Epoch|4/6 Epoch|5/6 Epoch|1 Epoch|
> |:-|:-:|:-:|:-:|:-:|:-:|:-:|
> |Dual-PPO|6.592±1.054|6.902±1.114|7.100±0.977|7.298±0.773|7.586±0.402|7.679±0.249|
> |DAR|6.742±0.062|7.870±0.041|8.160±0.024|8.301±0.035|8.353±0.046|8.369±0.019|
>
> These results demonstrate that DAR achieves substantially lower variance and more stable learning dynamics compared to Dual-PPO across both datasets.

---

> > ### Comment · Reviewer_GWzK · 2025-11-25
> >
> > Thanks for the additional experiments and clarifications. Wrt "Our claim of improved learning stability for DAR is supported by Figures 5(a) and 5(b)", what is the value of alpha? How is alpha tuned?
> > Do we need to tune alpha carefully or training is in general stable regardless of the choice of alpha?

---

> > > ### Author Response · Authors · 2025-11-26
> > >
> > > We thank the reviewer for the continued engagement during the discussion phase. We are pleased that our response has partially addressed your concerns.
> > >
> > > ### 1. What is the value of $\alpha$?
> > > For Figures 5(a) and 5(b), $\alpha = 0.1$ for DAR and DAO, while $\alpha = 0.3$ for Dual-PPO. The search range for $\alpha$ is [0.1, 0.3, 0.5, 0.7, 0.9]. Besides, we provide the full hyperparameter search range and the specific $\alpha$ values used for each method in Appendix D.3.
> > >
> > > ### 2. How is $\alpha$ tuned?
> > > We employ a two-stage tuning strategy for DAR: tune $\beta$ first, then tune $\alpha$.
> > >
> > > First, we fix $\alpha$ at 0.5 (equal importance between the two regularization targets, which empirically serves as a good starting point) and tune $\beta$ for the optimal total regularization. After identifying the optimal $\beta$ value, we vary $\alpha$ within the search range above to find its optimal value.
> > >
> > > This strategy connects well to our objective analysis in Section 4.3, where we express the alignment objective as weighted regression in Equation (4):
> > >
> > > $$\arg \max\_{\pi\_\theta} \mathbb{E}\_{x \sim \mathcal{D},y \sim \pi\_t(y|x)} \left(\frac{\pi\_{\text{0}}(y|x)}{\pi\_t(y|x)}\right)^{{\alpha}} \exp\left(\frac{1}{\beta}A(x,y)\right) \log\pi\_{\theta}(y|x)$$
> > >
> > > Theoretically, our "$\beta$ first, then $\alpha$" strategy first determines the scaling factor $\frac{1}{\beta}A(x,y)$ for the reward-related advantage information, then tunes $\alpha$ to control the discounting factor $\left(\frac{\pi_{\text{0}}(y|x)}{\pi_t(y|x)}\right)^{{\alpha}}$, which determines how conservative the policy should be toward responses with higher divergence from $\pi_0$.
> > >
> > > ### 3. Does $\alpha$ require careful tuning, or is training generally stable regardless of its value?
> > > Yes, careful tuning of $\alpha$ is crucial for achieving optimal performance.
> > >
> > > The RLHF framework is inherently sensitive to KL penalties, so varying KL coefficients (both $\beta$ and $\alpha$) significantly impacts alignment results. This sensitivity explains why the RLHF community now emphasizes Pareto analysis of reward/KL trade-offs when evaluating alignment algorithms.
> > >
> > > Our ablation study on $\alpha$ provides convincing evidence that the trade-off coefficient should be tuned carefully. In Figure 5(c), the results demonstrate that gradually decreasing $\alpha$ improves alignment performance, while the extreme case of $\alpha = 0$ leads to reward hacking. A balanced trade-off between reference regularization and stable optimization at $\alpha = 0.1$ yields optimal results.
> > >
> > > The trade-off coefficient $\alpha$ provides an additional dimension for specifying desired behavior that was previously inaccessible under overly restrictive regularization frameworks. When larger behavior changes are desired, we should encourage the policy to explore beyond the $\pi_0$ support region. Meanwhile, it is our transformation from a reward-maximizing objective into a weighted SFT loss ensures learning/alignment stability, which reduces seed-wise variance (as shown for PPO) and prevents training collapse (as shown for DAO).
> > >
> > > ---
> > >
> > > We sincerely appreciate your thoughtful engagement and the constructive nature of this discussion, which has genuinely helped us improve our work. If we have satisfactorily addressed your concerns, we would greatly appreciate your reevaluation of our submission. We remain available for any further clarifications you might need.

---

> > > > ### Author Response · Authors · 2025-11-28
> > > >
> > > > Dear Reviewer GWzK,
> > > >
> > > > Thank you very much for your previous response and for the additional questions.
> > > >
> > > > We appreciate your continued engagement with our work. We have addressed each of your follow-up questions above and believe these clarifications further strengthen the manuscript. Please kindly let us know if any points require further discussion.
> > > >
> > > > Thank you again for your valuable feedback.
> > > >
> > > > Sincerely,\
> > > > Authors

---

> ### Author Response · Authors · 2025-11-23
>
> ### Q4: Figure Caption Clarity
>
> We thank the reviewer for this suggestion. We have revised the caption to explicitly map colors/shapes to policies/regions and clearly define "search expansion".
>
> ### Q5: To provide end-to-end scripts including prompts, judge configs, seeds, filtering
> We appreciate this suggestion and are committed to full reproducibility. While we do not currently have packaged end-to-end scripts, we have included AI judge configurations, prompts, and experimental details in the appendix, along with our algorithm's source code in the supplement.
>
> We promise to open-source our complete codebase on GitHub, with the repository link provided in the camera-ready version. We view this open-source contribution as an important resource for the RLHF community to build upon our work.
>
> [1] Peng, Xue Bin, et al. "Advantage-weighted regression: Simple and scalable off-policy reinforcement learning." arXiv preprint arXiv:1910.00177 (2019).
>
> ---
>
> We thank the reviewer for these valuable feedback and hope these additional experiments and clarifications adequately address your concerns.

---

### Official Review · Reviewer_rHRn · 2025-10-31

**Soundness:** 2
**Presentation:** 3
**Contribution:** 2
**Rating:** 4
**Confidence:** 2

**Summary:**

This paper proposes a dual-KL regularization framework for RLHF that unifies two objectives usually treated separately: (1) preventing reward hacking via KL to the initial SFT model π₀, and (2) maintaining stability via KL or clipping to the current policy πₜ.
The authors show that these can be merged into a single interpolated reference in log-space, leading to a new weighted-SFT formulation called DAR (Dual-regularized Advantage Regression). DAR is positioned as a simple, RL-free alternative to PPO/GRPO, with theoretical analysis and experiments on Qwen2-7B showing improved reward–KL trade-offs.

**Strengths:**

1. Clear identification of a long-standing conflict between stability and reference regularization.
2. Mathematical formulation is elegant and internally consistent.
3. DAR simplifies PPO-style RLHF into a regression-like loss that is easier to implement and more stable.

**Weaknesses:**

1. Outdated baseline setup. All experiments use Qwen2-7B and compare mainly against PPO, GRPO, and RLOO; no comparison to modern alignment frameworks, stronger models, and new RL methods.
2. The novelty is mostly formal: the “dual-KL” is effectively a convex interpolation between π₀ and πₜ, similar to prior multi-reference ideas.
3. Theoretical results rely on clean advantage estimation; no analysis under noisy or biased rewards.
4. Empirical gains are modest and might vanish under stronger baselines.

**Questions:**

1.How sensitive is performance to α? Could an adaptive trade-off help?
2.Would DAR remain stable under noisy or AI-feedback reward models?

---

> ### Author Response · Authors · 2025-11-23
>
> We thank the reviewer for their feedback and positive comments on our motivation and algorithmic contribution. We have thoroughly examined your questions about our nolvelty and empirical studies, and would like to provide clarification on each point raised:
>
> ### Q1: Outdated baseline selection. Comparison to stronger models? Empirical gains are modest and might vanish under stronger baselines.
> We respectfully disagree with it. Our baselines span recent years:\
> 2019: PPO\
> 2023: DPO, SLiC \
> 2024: IPO, RLOO, SimPO, Online DAAs\
> 2025: GRPO
>
> To address concerns about stronger models, we additionally conducted new experiments using Qwen-2.5-7B as the base model, Qwen-3-32B as the LLM annotator, and GPT-5.1 as the evaluation judge. Results over three seeds show DAR consistently outperforms online RLHF methods across all three datasets:
>
> |Ref Win%|TL;DR|Helpful|Harmless|Mean|
> |:-|:-:|:-:|:-:|:-:|
> |PPO|60.70%±5.76%|61.73%±0.95%|72.50%±3.76%|64.98%|
> |RLOO|68.80%±1.48%|84.52%±0.54%|76.63%±1.64%|76.65%|
> |GRPO|68.27%±1.68%|77.72%±0.76%|79.63%±1.70%|75.21%|
> |**DAR**|**80.07%±0.64%**|**86.46%±0.54%**|**81.28%±0.40%**|**82.60%**|
>
>
> We welcome specific suggestions for additional online alignment methods to include in our comparison. This would help us further validate our approach.
>
>
> ### Q2: The novelty is mostly formal, the “dual-KL” is effectively a convex interpolation between π₀ and πₜ, similar to prior multi-reference ideas.
> We respectfully disagree. And our contribution extends beyond "form" in two key ways:
>
> **First, we address a previously unexplored problem in RLHF.** While our objective appears straightforward in unifying two regularization terms via dual-KL, the theoretical derivation showing it reduces to weighted supervised fine-tuning is novel and non-trivial.
>
> **Second, our motivation differs fundamentally from multi-reference methods.** Multi-reference methods [1] introduce additional reference models to improve diversity and lift single-reference constraints. In contrast, DAR explicitly addresses the inherent conflict between reference regularization and stable optimization through a principled trade-off mechanism. We resolve existing tensions in RLHF rather than augmenting it with additional components.
>
>
> ### Q3: Theoretical results rely on clean advantage estimation; no analysis under noisy or biased rewards.
> While prior works (RLOO [2], GRPO [3]) acknowledge noise in advantage estimation, none provides theoretical analysis of alignment performance under reward noise.
>
> In our paper, we addressed this empirically. In our ablation study in Figure 5(e), we progressively reduced Monte-Carlo sampling size from 16 to 1 to amplify advantage estimation noise. Results demonstrate that DAR exhibits the most robust alignment stability among all baselines, including GRPO, RLOO, and Iterative-SFT.
>
>
> ### Q4: How sensitive is performance to α? Could an adaptive trade-off help?
> Please refer to our ablation study on $\alpha$ in Figure 5(c). Our results demonstrate that tuning $\alpha$ balances conservatism (staying closer to $\pi_0$) versus exploitation (moving away from $\pi_0$ for higher win rates, with increased reward hacking risk). Only an appropriate trade-off ($\alpha=0.1$) between these two regularization can lead optimal alignment performance.
>
> An adaptive (or instance-based) trade-off coefficient could indeed provide further improvements, as the optimal trade-off likely varies across alignment process (or prompt-response pairs). We leave this promising direction to future work.
>
> ### Q5: Would DAR remain stable under noisy or AI-feedback reward models?
> Yes, DAR remains stable under noisy AI-feedback. When we eliminate Monte-Carlo sampling (sampling size = 1), our results in Figure 5(e) show that DAR maintains near-optimal performance with superior robustness compared to baseline methods.
>
>
> [1] Aminian, Gholamali, et al. "KL-Regularized RLHF with Multiple Reference Models: Exact Solutions and Sample Complexity." The Thirty-ninth Annual Conference on Neural Information Processing Systems. 2025.\
> [2] Ahmadian, Arash, et al. "Back to basics: Revisiting reinforce style optimization for learning from human feedback in llms." arXiv preprint arXiv:2402.14740 (2024).\
> [3] Shao, Zhihong, et al. "Deepseekmath: Pushing the limits of mathematical reasoning in open language models." arXiv preprint arXiv:2402.03300 (2024).
>
> ---
> We sincerely appreciate your constructive feedback, which has helped us improve the quality and clarity of our paper.

---

> > ### Author Response · Authors · 2025-11-28
> >
> > Dear Reviewer rHRn,
> >
> > As we are now in the last week of the discussion phase, we would greatly appreciate your feedback on whether we have appropriately addressed your concerns in our rebuttal and revised manuscript. If you have any remaining concerns or would like further clarification, please kindly let us know, and we will do our best to resolve them.
> >
> > Thank you for your time and consideration.
> >
> > Sincerely,\
> > Authors

---

### Official Review · Reviewer_7Noe · 2025-10-31

**Soundness:** 2
**Presentation:** 3
**Contribution:** 3
**Rating:** 6
**Confidence:** 4

**Summary:**

The paper proposes an approach to address the trade-offs arising from simultaneously regularizing towards the reference policy (to mitigate reward hacking) and the current policy (for stable policy updates), in RLHF. They accomplish this by regularizing towards a convex combination of the reference policy and current policy. This is done via a weighted supervised fine-tuning loss, which allows for stable training. Experimentally, their proposed approach improves over baselines (both online RL-based approaches and online/offline direct alignment algorithms)

**Strengths:**

The paper addresses a critical problem that has not been explored in the literature, the impact of regularizing towards both the reference policy and the current policy, in RLHF. Regularizing towards the reference policy is done by a KL penalty to the reward, and regulazing towards the current policy is achieved via clipping or a KL constraint. Together, these two constraint our objective to operate in the intersection of the trust region that becomes increasingly restrictive as training progresses. The paper proposes a simple weighted supervised fine-tuning objective by regularizing towards a convex combination of both the reference policy and the current policy, leading to a stable training algorithm DAR (Dual-Regularized Advantage Regression). Experimental results showcase that DAR surpasses online RL based methods (PPO, GRPO, RLOO) and online/offline direct alignment methods (DPO/IPO/SLiC) across three training domains. The presentation is clear, crisp with no major issues with the grammar and writing flow.

**Weaknesses:**

One of my concerns with the paper is their choice to regularize towards a convex combination of the reference policy $\pi_{0}$ and the current policy $\pi_{t}$ i.e $\alpha D_{KL}(\pi \vert\vert \pi_{0}) + (1-\alpha) D_{KL}(\pi \vert\vert \pi_{t})$. This inherently leads to incentivizing regularizing to one of the distibutions than the other (when $\alpha$ != 0.5). It would have been better to have two independent multipliers for each of the divergence, which supports the Lagrangian view of the objective when looking at the divergences as constraints i.e $\alpha_{1} D_{KL}(\pi \vert\vert \pi_{0}) + \alpha_{2} D_{KL}(\pi \vert\vert \pi_{t})$.

Additionally, in proposition 4.1, the objective has a KL penalty wrt a reference mixture distribution $\pi_{ref} = \pi_{0}^{\alpha}\pi_{t}^{1-\alpha}$. $\pi_{ref}$ need not be a valid probability distribution, since it may not sum up to 1. The KL term here hence may not be between two distributions. There needs to be a normalizing factor for $\pi_{ref}$ to make it a valid probability distribution. This would affect their proof of the optimal policy in Theorem 4.2, since there would now be normalizing factors of $\pi_{ref}$ in the expressions.

In the DAR derivation in Appendix C.3, ignoring the earlier issue with $\pi_{ref}$ not being normalized, on line 885, they state "we factor out the partition function $Z(x)$ as it is a positive constant and doesnt shift optimal policy". $Z(x)$ depends on $x$ and leads to weighing each prompt differently. Considering a simple setting with two possible $x$, the objective is $Z(x_{1}) f_{\theta}(x_{1}) + Z(x_{2}) f_{\theta}(x_{2})$.  How can $Z(x)$ be factored out without affecting the objective

The authors state that as $\pi_{t}$ is trained, the log-likelihood interpolation constructs a reference target that is inherently positioned closer to the optimal policy. No proof for this is provided. If this is solely because $\pi_{ref}$ contains $\pi_{t}^{1-\alpha}$, that becomes more optimal over the course of training. If so, standard PPO also has a policy constraint with respect to $\pi_{t}$. How is yours more optimal?

**Questions:**

1) Is there a reason for choosing a convex combination of the two KL divergences for DAR, instead of choosing independent multipliers?

2) Why is the $\pi_{ref}$ not normalized in the KL constraint for DAR and does this impact the derivation of the optimal policy for DAR?

3) Why can $Z(x)$ be dropped from the objective, without modifying it, in the DAR derivation in Appendix C.3?

4) Is there a theoretical justification for this statement "as $\pi_{t}$ is trained, the log-likelihood interpolation constructs a reference target that is inherently positioned closer to the optimal policy." in line 240?


Minor Questions/Nits

- Equation 2 operates at trajectory level, whereas PPO is defined at token-level
- What the reward (avg, total, etc) in Table 1. Explain it in detail in caption.
- Why no comaprision against PPO in the the Experiments for Standard RLHF (line 317)?
- In Figure 3, why no evolution plots for online direct alignment methods?
- Improvements on MT-Bench and AlpacaEval 2.0 in Table 3 seem marginal. Also the column is named "AlphacaEval".

---

> ### Author Response · Authors · 2025-11-23
>
> We appreciate the reviewer's recognition of our motivation and presentation. We have carefully considered your concerns about the dual-constrained objective and respond to each point below:
>
> ### Q1: Why regularize towards a convex combination instead of two independent multipliers?
> There are two main reasons for using a convex combination:
> 1. Regularizing towards two targets simultaneously inherently involves a trade-off. A convex combination naturally captures this competing relationship and allows us to assign relative importance to each term given a fixed total regularization strength.
> 2. The convex combination aligns better with our final weighted SFT loss. Specifically, α controls the discounting factor in the regularization weight, while β scales the advantage value in the advantage weight. This unified structure maintains consistency throughout our framework.
>
> ### Q2: The normalizing factor in proposition 4.1.
> Thank you for this valuable comment. We acknowledge that the unified reference target should indeed include a normalizing factor,
> $$\\max_{\\pi_\\theta} \\mathbb{E}\_{x\\sim D, y\\sim \\pi_\\theta}[A(x,y)] -\\beta \\mathbb{D}\_\\text{KL}(\\pi_\\theta(y|x)|| \\frac{1}{C(x)}  \\pi_0(y|x)^\\alpha  \\pi\_{t}(y|x)^{1-\\alpha}),$$ where $C(x) = \sum\_y \pi_0(y|x)^\alpha \pi_t(y|x)^{1-\alpha}$. We have corrected this in the updated version.
>
> We have then re-verified Theorem 4.1 based on the updated proposition. The added normalizing factor in the end turns into $\max\_{\pi_\theta} \mathbb{E}\_{x\sim D, y\sim \pi\_\theta}\beta \log C(x)$ as constant independent on $\pi_\theta$ can can be removed from the objective. Thus, the theorem remains unchanged.
>
>
> ### Q3: Regarding the partition function in the DAR's derivation
> The removal of $Z(x)$ in our derivation is mathematically valid and follows the same approach used in previous work of Advantage Weighted Regression [1].
>
> This is to say:
> $$ \arg \max\_{\pi_\theta} \mathbb{E}\_{x\sim D, y\sim \pi\_t(y|x)} \bigg(\frac{\pi\_0(y|x)}{\pi\_t(y|x)}\bigg)^\alpha \exp\big(\frac{1}{\beta}A(x,y)\big) \log \pi\_\theta(y|x) =\arg\max\_{\pi\_\theta} \mathbb{E}\_{x\sim D, y\sim \pi\_t(y|x)} \frac{1}{Z(x)} \bigg(\frac{\pi\_0(y|x)}{\pi\_t(y|x)}\bigg)^\alpha \exp\big(\frac{1}{\beta}A(x,y)\big) \log \pi\_\theta(y|x)$$ where $\frac{1}{Z(x)}>0$ for all prompts.
>
> Since for each prompt $x$, the partition function $Z(x)$ is just a positive scaling factor that does not depend on y. So the policy that maximizes the scaled objective is exactly the same as the one maximizes the unscaled objective. In other words, this scaling factor does not affect optimal responses y* (for each prompt x) receives the highest weight. Therefore, dropping this positive constant does not alter the underlying optimal policy.
>
>
> ### Q4: How does the dual-constrained objective provide better interpolation than standard PPO with a policy constraint w.r.t. $\pi_t$?
>
> Thank you for this question. Here we provide a proof:
>
> At each iteration, we minimize the KL divergence towards the optimal policy:
> $$\min\_{\pi_\theta} \mathbb{D}\_\text{KL}(\pi^*||\pi\_\theta)$$
>
> With a gradient update $\pi\_{t+1} = \pi\_t - \eta \cdot \nabla\_{\pi\_\theta} \mathbb{D}\_\text{KL}(\pi^*||\pi\_\theta){|}\_{\pi\_\theta=\pi\_t}$. We have:
>
> $$\mathbb{D}\_\text{KL}(\pi^* | \pi_{t+1}) \\leq \mathbb{D}\_\text{KL}(\pi^* | \pi_{t})$$
>
> This implies that our effective reference target also improves:
>
> $$\mathbb{D}\_\text{KL}\left(\pi^*|| \frac{1}{C(x;\pi_0,\pi\_{t+1})} \pi\_{0}(y|x)^{\alpha} \pi\_{t+1}(y|x)^{1-\alpha}\right) \\leq \\mathbb{D}\_\\text{KL}\left( \pi^\*||  \\frac{1}{C(x;\\pi\_0,\\pi\_t)} \\pi\_{0}(y|x)^{\\alpha} \\pi\_t(y|x)^{1-\alpha}\right)$$
>
> where $C(x;\pi_0,\pi_t) = \sum_y \pi_0(y|x)^\alpha \pi_t(y|x)^{1-\alpha}$ is the normalizing partition function.
>
> **Proof sketch:** Assuming a sufficiently small learning rate ($\eta$), the partition function changes minimally between updates $C(x;\pi_0,\pi_{t+1}) \approx C(x;\pi_0,\pi_t) $ . Therefore, the KL divergence from the effective reference target to the optimal policy as $\\mathbb{D}\_\\text{KL}\\left(\\pi^\*|| \\frac{1}{C(x;\\pi\_0,\\pi\_t)} \\pi\_{0}(y|x)^{\\alpha} \\pi\_t(y|x)^{1-\alpha}\right)$ is primarily determined by the dynamic component $\pi\_t$, that is $\\mathbb{E}\_{y \sim \pi^\*} (1-\alpha) \log \frac{\pi^\*(y|x)}{\pi\_t(y|x)}$, where the rest remains invariant to either $t$ or $t+1$. Since $\pi\_{t+1}$ is closer to $\pi^\*$ than $\pi_t$ in KL divergence, the effective reference target exhibits monotonically decreasing KL divergence to $\pi^*$ as $\pi_t$ progressively aligns with the optimal policy.

---

> ### Author Response · Authors · 2025-11-23
>
> **Key advantage over PPO:** While PPO constrains the policy to remain close to a fixed reference $\pi_0$, our dual-constrained objective (with flexible trade-off between the two KL) maintains regularization towards a dynamically improving target that blends $\pi_0$ and the current policy $\pi_t$. This allows our method to benefit from continuous improvement via the evolving $\pi_t$ component.
>
> ### Q5: Minior Questions
> 1. PPO definition: Corrected to specify token-level operation.
> 2. Table 1 rewards: Clarified in the updated caption that these are average rewards.
> 3. PPO on Helpsteer2: We excluded PPO from Helpsteer2 for practical reasons. Despite extensive experiments, we could not identify any PPO configuration (using a 1.5B value model) that consistently outperform the strong base model across all three metrics. The root cause is the significant capacity mismatch between 1.5B value model and 7B policy model. However, we are already operating at our computational limits (fine-tuning a 7B policy model while querying a 72B reward model). We therefore use Monte-Carlo based methods (RLOO and GRPO) as baselines for Helpsteer2. These methods do not require a separate value model, and notably, Monte-Carlo approaches achieved the best alignment performance in the original Helpsteer2 paper [2], making them well-suited baselines for our experiments.
> 4. Win rate curves of online direct alignment method: we plot separately in Figure 6 in the appendix for better presentation purpose, as they uses more epoch to train.
> 5. MT-Bench and Alpaca Eval: thank you for pointing out this typo, which is now corrected. We acknowledge that our method's advantage depends on the expected scale of policy shift. On the HelpSteer2 dataset, since the reference policy is the well-tuned Qwen2-7B-Instruct, we intuitively expect smaller policy alignment shift, making the conflict between $\pi_0$ and $\pi_t$ less pronounced than in the helpful and TL;DR datasets. Meanwhile, our empirical results show that even in these cases, our method still provides better alignment results, demonstrating the robustness of our algorithm.
>
> [1] Peng, Xue Bin, et al. "Advantage-weighted regression: Simple and scalable off-policy reinforcement learning." arXiv preprint arXiv:1910.00177 (2019).\
> [2] Wang, Zhilin, et al. "Helpsteer2-preference: Complementing ratings with preferences." arXiv preprint arXiv:2410.01257 (2024).
>
> ---
> Thank you again for your constructive feedback, which has helped us improve the quality and clarity of our paper.

---

> > ### Author Response · Authors · 2025-11-28
> >
> > Dear Reviewer 7Noe,
> >
> > As we are now in the last week of the discussion phase, we would greatly appreciate your feedback on whether we have appropriately addressed your concerns in our rebuttal and revised manuscript. If you have any remaining concerns or would like further clarification, please kindly let us know, and we will do our best to resolve them.
> >
> > Thank you for your time and consideration.
> >
> > Sincerely,
> > Authors

---

### Author Response · Authors · 2025-12-02

Dear Area Chair and Senior Area Chair,

We sincerely thank you and all reviewers for the significant time and effort dedicated to evaluating our work.

To facilitate the meta-review process, we briefly summarize our paper's contributions, the author-reviewer discussions, and the revisions made during rebuttal.

**1. Summary of Contributions**

This paper addresses a previously unstudied problem in PPO-based RLHF methods: the overly restrictive regularization framework that arises from conflicts between stable optimization and reference regularization. We propose to unify these two regularization mechanisms as dual-KL penalties, with a convex combination coefficient enabling flexible trade-offs between them. Our practical algorithm, DAR, optimizes a stable weighted SFT loss derived from the dual-KL alignment objective. Our comprehensive experiments demonstrate that DAR achieves better alignment results than online RLHF methods (through dual-KL alignment) while maintaining enhanced stability compared to dual-KL based RL algorithms (through conversion to weighted SFT).

**2. Rebuttal Summary**

During the discussion stage, Reviewers 4Fmj and GWzK actively engaged with us. **Reviewer 4Fmj (score 6)** explicitly confirmed that our rebuttal addressed their questions. Besides, we believe we have also resolved **Reviewer GWzK's (score 4)** concerns, as they acknowledged our additional experiments and clarifications, and we properly addressed their follow-up questions about technical details.

Although Reviewers 7Noe and rHRn were unable to respond before the cutoff, we fully addressed their concerns:
* **Reviewer 7Noe (score 6)** provided positive feedback while raising theoretical concerns. Our rebuttal directly addressed these by clarifying the derivation of DAR, the advantages of the interpolated reference target, and the design considerations behind the trade-off coefficient.
* **Reviewer rHRn (score 4)** acknowledged our algorithmic contributions (*"DAR simplifies PPO-style RLHF... easier to implement and more stable"*, *"Mathematical formulation is elegant and internally consistent"*), but requested further validation experiments with newer models. Our rebuttal provided the requested empirical evidence and clarified our work's novelty, particularly relative to the recent multi-reference RLHF approach [1].

[1] Aminian, Gholamali, et al. "Theoretical Analysis of KL-regularized RLHF with Multiple Reference Models." arXiv preprint arXiv:2502.01203 (2025).

**3. Revision Summary**

* Updated the unified reference target to include a normalizing factor (Reviewers 7Noe and 4Fmj)
* Conducted additional experiments using Qwen2.5-7B (newer backbone), Qwen3-32B (smaller LLM annotator with noisier labels), and GPT-5.1 (SOTA LLM as external judge), which validate our method's effectiveness through clear outperformance over online RLHF baselines (Reviewers rHRn and GWzK)
* Added ablation study comparing stable optimization in forward KL vs. reverse KL form (Reviewer GWzK)
* Reorganized method section to introduce the dual-KL alignment objective before empirical validation of Dual-PPO vs. PPO (Reviewer 4Fmj)

We once again thank you for your time and support throughout the review process, your dedication truly advances our field.

Best regards,\
The Authors

---

### Meta-Review · Area_Chair_KeTc · 2026-01-12

**Summary:**

Summary of the reviewers' concerns:

- Why did the authors choose a convex combination of the two KL divergences for DAR, instead of choosing independent multipliers?

- Why is the $\pi\_{\mathrm{ref}}$ not normalized in the KL constraint for DAR and does this impact the derivation of the optimal policy for DAR?

- Why can $Z(x)$ be dropped from the objective, without modifying it, in the DAR derivation in Appendix C.3?

- Is there a theoretical justification for this statement "as $\pi\_t$ is trained, the log-likelihood interpolation constructs a reference target that is inherently positioned closer to the optimal policy." in line 240?

- Outdated baseline setup: All experiments use Qwen2 and Llama-3.1 and compare mainly against PPO, GRPO, and RLOO. Moreover, Figure 3/Table 2 use GPT-4 Turbo as the judge and Qwen2-72B-Instruct as the LLM annotator. Results would be stronger with a stronger contemporaneous judge (e.g., GPT-5) and annotator (Qwen3) for additional validation.

- The novelty is mostly formal: the "dual-KL" is effectively a convex interpolation between $\pi\_0$ and $\pi\_t$, similar to prior multi-reference ideas.

- Theoretical results rely on clean advantage estimation; no analysis under noisy or biased rewards.
Empirical gains are modest and might vanish under stronger baselines.

- How sensitive is performance to $\alpha$? Could an adaptive trade-off help?

- Would DAR remain stable under noisy or AI-feedback reward models?

- In the discussion of PPO stability, the classical TRPO/PPO literature typically regularizes with a KL of the form $D_{\mathrm{KL}}\bigl[\pi\_{\text{old}} \,\|\, \pi\_{\theta}\bigr]$. By contrast, the paper's PPO-Align constrains with $D_{\mathrm{KL}}\bigl[\pi\_{\theta} \,\|\, \pi\_{t}\bigr]$ and penalizes $D_{\mathrm{KL}}\bigl[\pi\_{\theta} \,\|\, \pi\_{0}\bigr]$. The rationale for changing directions relative to the classical trust-region view is not made explicit.

- While DAR (weighted SFT) is argued to be stable, quantitative stability analyses for the dual-PPO path are sparse.

- Is Eq (2) PPO for RL not RLHF, since there is no KL penalty to $\pi\_{\mathrm{ref}}$? The equation in Section 4.1 seems to transform the KL penalty in Eq (1) into a hard constraint $KL < \epsilon$.

- The dual-KL (Eq. 3) looks like a special case of \cite{aminian2025multi} with 2 references. What are the differences and additional contributions?

- In Proposition 4.1, $\log \pi\_{0}(y|x) = \alpha \log \pi\_{0}(y|x) + (1-\alpha)\log \pi\_{t}(y|x) + C(x)$ with a constant normalization $C(x)$ is suggested to ensure policies summing up to $1$.

- What are the evaluation metrics of MT Bench in Table 3?

- In Figure 5c, could you explain more about EOS-missing rate, and why $\alpha = 0$ increases EOS-missing rate? Is it convenient to add LC-win rate?

- In Algorithm 1, what are $\mu\_A$ and $\sigma\_A$? What are the meanings of the two weights $w^{i}\_{\mathrm{reg}}$ and $w^{i}\_{\mathrm{adv}}$ and how are they computed?

**Reviewer Concerns:**

The rebuttal is thorough and effectively addresses all the reviewers' major concerns. The authors have provided new empirical results and detailed clarification in the rebuttal for all reviewers' questions. The authors have also incorporated the corresponding revisions into the paper. Overall, this work has made a good contribution to the field of RLHF.

**Reviewer Scores:**

Reviewer 4Fmj stated that the concerns have been addressed and the rating of 6 will be kept. Reviewer 7Noe has an initial rating of 6, and I think Reviewer 7Noe's concerns have been resolved as well. Reviewers GWzK and rHRn have initial ratings of 4. As their concerns are well addressed, I expect the two reviewers will raise their ratings accordingly.

---

### Decision · Program_Chairs · 2026-01-26

Accept (Poster)